

# Smart grid energy scheduling based on improved dynamic programming algorithm and LSTM

Xiaoyu Huang, Yubin Lin, Xiaofei Ruan, Jiyu Li and Nuo Cheng

Department of Evaluation Center, Economic and Technological Research Institute of State Grid Fujian Electric Power Co., Ltd, Fuzhou, Fujian, China

## ABSTRACT

The optimal scheduling of energy in a smart grid is crucial to the energy consumption of the entire grid. In fact, for larger grids, intelligent scheduling may result in substantial energy savings. Herein, we introduce an enhanced dynamic programming algorithm (DPA) that utilizes two state variables to derive the optimal power supply schedule. The algorithm accounts for the dynamic states of both batteries and supercapacitors in the power supply system to augment the performance of the dynamic programming model. Additionally, this study incorporates a long short-term memory (LSTM) deep learning model, which integrates various environmental factors such as temperature, humidity, wind, and precipitation to predict grid power consumption. This serves as a mid-point pre-processing step for smart grid energy consumption scheduling. Our simulation experiments confirm that the proposed method significantly reduces energy consumption, surpassing similar grid energy consumption scheduling algorithms. This is critical for the establishment of smart grids and the reduction of energy consumption and emissions.

## INTRODUCTION

With the continuous evolution of the electric power industry, the conventional power grid system is no longer suited for the current stage of power demand. Power supply enterprises must now ensure not only adequate but also high-quality power supply (*Tuballa & Abundo, 2016*). Power system dispatching has long been a topic of concern for scholars in the electric power field, with the main aim of achieving minimal energy consumption and economic cost while meeting various system conditions (*Butt, Zulqarnain & Butt, 2021*). A smart grid is built on the basis of a integrated, high-speed two-way communication network, through advanced sensing and measurement technology, advanced equipment technology, advanced control for methods as well as an advanced decision support control power grid system. Particularly in the current global low-carbon environment, reducing energy consumption in power grid systems to decrease fossil fuel consumption and ensure high-quality electricity supply is an issue that must be addressed (*Dileep, 2020*).

The earliest power system energy scheduling problems were solved using traditional mathematical methods. For instance, *Lee, Park & Ortiz (1984)* devised a fuel cost

Corresponding author
Xiaoyu Huang,
15060087102@163.com

formulation for optimal real and reactive power scheduling to enable economic operation of power systems. The problem was decomposed into p-optimized and q-optimized modules, which employed the same fuel cost objective function to obtain the optimal solution. *Jan & Chen (1995)* applied the sensitivity factorization method to the power system economic dispatch problem and combined it with fast Newton–Raphson economic dispatch to solve the optimal power allocation problem. *Sun (1993)* proposed an optimization algorithm based on consistency and gradient descent to minimize economic cost. The method uses a distributed strategy to transfer information only between interconnected buses without passing all the information to all the buses, which extremely balances the supply-side and demand-side power. Setting the proper step size and initial value allows the algorithm to converge to the optimal solution quickly. However, these traditional mathematics-based methods are limited by their dependence on manual mathematical modeling to solve the problem. The more intricate the grid model, the more complex mathematical modeling is required to complete it, and the model constructed is not flexible enough to adapt to various grid systems.

## RELATED WORKS

As the electric power industry continues to grow, power generation systems have become increasingly complex, with nonlinearity, high dimensionality, and intricate constraints. Recently, computer and artificial intelligence techniques have been introduced to automate the solution of some of these complex problems (*Omitaomu & Niu, 2021*). Various machine learning (*Janiesch, Zschech & Heinrich, 2021*) methods have been used to solve the scheduling problems of complex power grids, including DPAs (*Lui et al., 2020*), particle swarm algorithms (*Shami et al., 2022*), ant colony algorithms (*Luo et al., 2020*), and random forests (*Rigatti, 2017*), among others. *Qin et al. (2017)* proposed an improved orthogonal design particle swarm optimization algorithm (IODPSO) for solving single- and multi-area economic load dispatch problems with generator nonlinearities, such as valve point effects, no-operating zones, ramp speed limits, and multiple fuels. *Pradhan, Roy & Pal (2016)* proposed the evolutionary optimization algorithm, Gray Wolf Optimization (GWO), based on gray wolf behavior while considering the nonlinear characteristics of generators. *Jayabarathi et al. (2016)* proposed an optimization algorithm based on consistency and gradient descent to find the minimum economic cost, which uses a distributed strategy to transfer information only between interconnected busbars, balancing the power on the supply and demand sides. *Xie et al. (2019)* proposed an improved two-stage compensated stochastic optimization algorithm based on recursive dynamic regression, which considers the high-dimensional correlation of multiple wind farms, and can overcome the difficulties of traditional stochastic optimization methods to solve the dynamic economic scheduling problem of high-dimensional correlation of multiple wind farms. While these smart optimization algorithms can deal with the energy consumption scheduling problem in smart grids to some extent, they do not consider the influence of external factors on grid energy consumption. Therefore, optimizing energy scheduling in the context of modern smart grids is the key problem that this article aims to address.

The problem of optimal scheduling of energy consumption in smart grids is of great significance for improving grid efficiency and conserving fossil energy. With the development of artificial intelligence, many intelligent algorithms are currently being used to solve the energy dispatching problem in power grids. However, traditional mathematics-based methods and some simple intelligent algorithms (*Hossain et al., 2019*) are not able to cope with the current complex grid environment.

Most machine learning algorithms for intelligent scheduling do not consider power consumption forecasting. Electricity consumption prediction parameters can be used to provide guidance for intelligent scheduling of electricity consumption to greatly reduce energy consumption losses. Time series prediction models are a class of models used in deep learning to solve similar problems. RNN (*Wojciech & Ilya, 2014*), as the earliest time series model, is widely used to solve problems such as natural language processing. However, due to the inability of RNNs to solve the long dependence difficulty in time series problems, LSTM (*Graves, 2012*) models came into being. Abbasimehr et al. (*Abbasimehr, Shabani & Tousefi, 2020*) implemented demand forecasting for businesses using an LSTM model, which uses a grid search approach to automatically select the best forecasting model for a given time series, considering different combinations of LSTM hyperparameters. Ma et al. (*Ma & Mao, 2020*) implemented industrial supplies lifetime forecasting using an LSTM model. These methods improve the LSTM model to varying degrees, however, they do not consider multiple influencing factors. Many methods take into account multiple state information when forecasting and analyzing stocks using time series models, and use this same data as inputs to the model, with the output only for stock movements. The inspiration from these works is that we can consider multiple factors that influence grid energy consumption and integrate these factors to more accurately predict grid electricity consumption. This would enable us to do the preliminary work of smart grid energy dispatch.

The problem is solved through the implementation of a DPA, which involves the subdivision of the problem into defined states and the establishment of relationships between these states. Over the years, numerous scholars have refined this algorithm to accommodate a diverse range of problem domains. *Fares et al. (2015)*, for instance, utilized the DPA to optimize fuel cells for hybrid vehicles. By allocating engine power optimally, they were able to significantly reduce pollution emissions. To expedite the convergence rate of the original DPA and improve its efficiency, the scholars incorporated weights into the fitness function. *Feng et al. (2017)* also leveraged a DPA to optimize hydropower systems, thereby achieving dimensionality reduction. They began by utilizing a uniform design to construct the set of state variables for each cycle, by selecting small yet representative combinations of discrete states. Next, they applied the DP recursive equation to obtain an enhanced solution for the next computational cycle. In another study, *Deng, Santos & Curran (2020)* optimized the aircraft maintenance inspection process using a DPA, which reduced the time between two aircraft maintenance procedures, thereby minimizing economic and time costs. *Wang, Kang & Liu (2020)*, on the other hand, proposed a model consisting of three components. Firstly, they suggested a dynamic programming model to solve the scheduling issue of electric bus fleets. Secondly, they proposed an inverse order

matching strategy that accounts for both the battery capacity degradation process and workload differences. Finally, they used a battery capacity degradation model to determine the amount of battery capacity loss and the number of battery replacements. For the energy consumption scheduling of the smart grid, it is worth noting that most of the improvements of the aforementioned methods lie in the algorithmic process, with minimal consideration of the state factor, which is especially crucial for the energy consumption scheduling of the smart grid.

In this article, we propose a smart grid energy scheduling algorithm based on the improved DPA algorithm and LSTM. The specific contributions are as follows:

(1) Environmental variables, including temperature, humidity, wind and precipitation, are used to determine the characteristics of power grid energy consumption.

(2) LSTM is used to predict power grid energy consumption as one of the conditions of energy scheduling.

(3) An improved DPA algorithm with two state variables is proposed, which lists both battery and supercapacitor as state variables in the smart grid.

## ENERGY CONSUMPTION SCHEDULING BASED ON LSTM AND IMPROVED DPA

To address the energy consumption scheduling problem in smart grids and maximize energy savings, this study proposes an energy consumption scheduling method that employs an enhanced DPA. However, it is crucial to note that the entire energy consumption scheduling process is not solely executed using the DPA. Rather, it relies on the accurate prediction of electricity consumption by a node in the future period. Figure 1 illustrates the overall method structure, which features a triangular configuration. In this configuration, the grid offers the fundamental data to the LSTM model for dataset construction and training. Once trained, the LSTM model predicts short-term energy consumption, which is then fed to the improved DPA. The DPA, which has been modeled and constructed in this study for grid energy consumption scheduling, produces the final scheduling outcomes that are sent to the grid for execution. LSTM provides the data representation after feature extraction, which is used as one of the bases of dynamic programming by DPA algorithm, and finally regulates the grid nodes.

### LSTM algorithm considering environmental factors

LSTM has found extensive application in natural language processing, where it addresses the long dependency problem in model training compared to RNN. The LSTM process is roughly divided into three steps. The first step is to determine what information to discard from the cell state, which is done through the oblivion gate. The second step is to determine what information is stored in the cell state, which is determined by the input gate. Finally, determine which information to output, which is determined by the output gate result and the cell state. However, to enable accurate prediction of energy consumption of a node in the power grid, LSTM requires ample data and an appropriate experimental setup. In reality, the variation of grid energy consumption is not only linked to the load but also to the environment where the grid is situated. To this end, this study collects the daily average

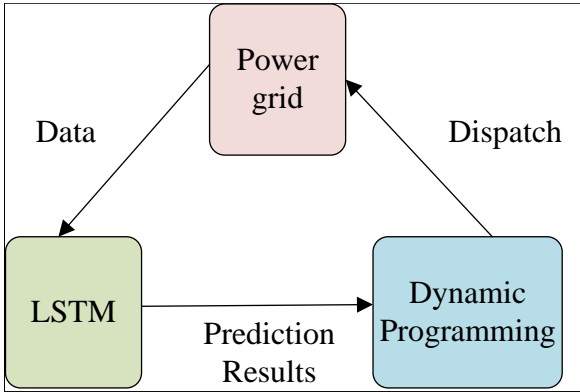

**Figure 1  Structure of energy consumption scheduling system.** The overall method structure, which features a triangular configuration, is illustrated. In this configuration, the grid offers the fundamental data to the LSTM model for dataset construction and training. Once trained, the LSTM model predicts short-term energy consumption, which is then fed to the improved DPA.

| Electricity consumption | Temperature | Wind velocity | Relative Humidity | Precipitation |
|---|---|---|---|---|
| Data 1 | Data 1 | Data 1 | Data 1 | Data 1 |
| ... | ... | ... | ... | ... |
| Data n | Data n | Data n | Data n | Data n |

**Figure 2  Input data structure.** This study collects the daily average data of wind speed, precipitation, temperature, and humidity in a city and trains them simultaneously with daily electricity consumption. Specifically, wind speed, precipitation, temperature, and humidity are input as influencing features of electricity consumption to the LSTM model. The input feature structure is shown in the figure.

data of wind speed, precipitation, temperature, and humidity in a city and trains them simultaneously with daily electricity consumption. Specifically, wind speed, precipitation, temperature, and humidity are input as influencing features of electricity consumption to the LSTM model. In this article, four kinds of environmental data of T City in 2020 and data of a node of the power grid are collected and organized in the form of matrix. The input feature structure is depicted in Fig. 2.

## Modeling of smart grid energy storage systems

The initial step towards addressing the smart grid energy consumption dispatching issue through dynamic programming is to develop a model for the energy storage system of the smart grid. The fundamental configuration of the smart grid energy storage system

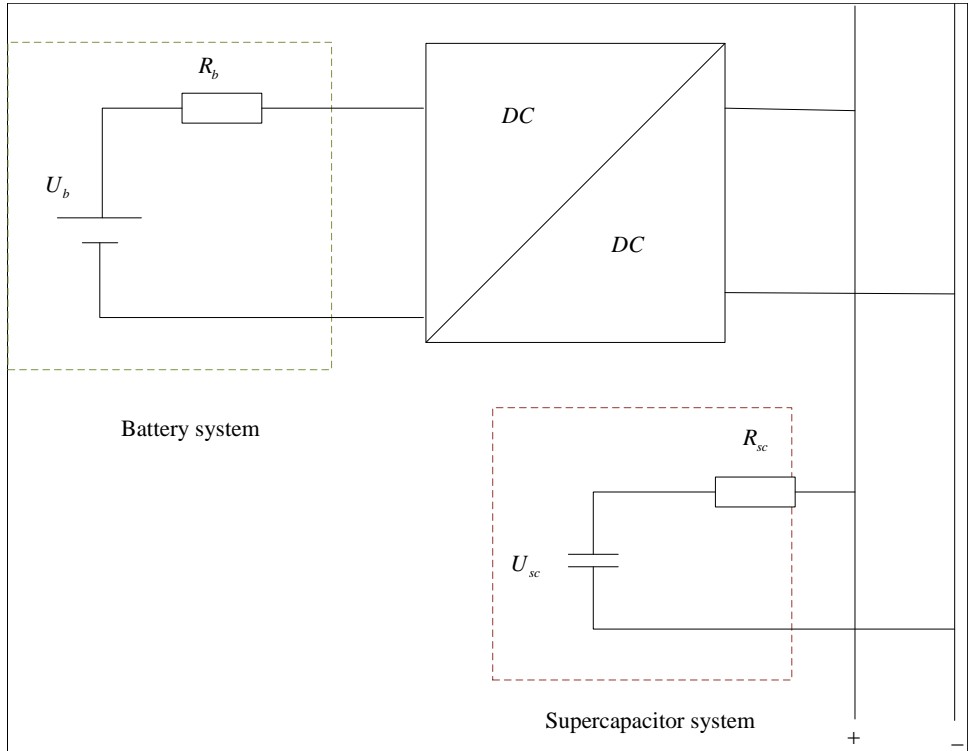

**Figure 3   Schematic diagram of smart grid energy storage system.** The fundamental configuration of the smart grid energy storage system is shown. The energy storage system of the smart grid, studied in this article, comprises primarily three components: the battery system, supercapacitor system, and DC/DC DC converter. Among these, the battery system is connected in parallel to the DC bus *via* the DC converter. The Rint model is used for the battery, and the internal resistance model is employed for the supercapacitor, with the internal resistance of both remaining constant.

is displayed in Fig. 3. The energy storage system of the smart grid, studied in this article, comprises primarily three components: the battery system, supercapacitor system, and DC/DC DC converter. Among these, the battery system is connected in parallel to the DC bus *via* the DC converter. The Rint model is used for the battery, and the internal resistance model is employed for the supercapacitor, with the internal resistance of both remaining constant. In conventional states, single state variables can describe state characteristics, but in smart power grids, the states of batteries and supercapacitors are equally important, so single state variables cannot fully describe their changes.

Assuming that the power demand of a node of the smart grid at the $k$ discrete event interval is $P_{req}(k)$, then the energy flow relationship of its on-board hybrid energy storage system is shown in Eq. (1).

$$P_{req}(k) = \begin{cases} \left(I_b(k)U_b(k) - I_b^2(k)\eta + I_{sc}(k)U_{sc}(k) - I_{sc}^2(k)R_{sc} \right. & P_{req}(k) > 0 \\ \dfrac{I_b(k)U_b(k) + I_b^2(k)R_b}{\eta} + I_{sc}(k)U_{sc}(k) + I_{sc}^2(k)R_{sc} & P_{req}(k) < 0 \end{cases} \tag{1}$$

where $U_b(k)$ is the battery no-load voltage, $I_b(k)$ is the battery current, $U_{sc}(k)$ is the supercapacitor no-load voltage, $I_{sc}(k)$ is the supercapacitor current; $\eta$ is the efficiency

of the DC converter; $P_{req}(k) > 0$ indicates the smart grid in operating state; $P_{req}(k) < 0$ indicates the smart grid in energy-saving state.

Equation (1) indicates that during normal operation mode of the smart grid, the system loss comprises two major components: the internal resistance thermal loss of the battery and supercapacitor, and the DC/DC DC converter loss.

The no-load voltage of the battery is considered as a function of the battery charge state ($SOC$), and the following functional relationship is obtained by conducting experiments on the hybrid power pulse characteristics of the battery monomer, performing parameter discrimination and fourth-order polynomial fitting:

$$U_b(k) = aSOC_b^4(k) + bSOC_b^3(k) + cSOC_b^2(k) + dSOC_b(k) + e \tag{2}$$

where, $a \sim e$ is the polynomial fit coefficient; $SOC_b(k)$ is the charge state of the battery at the $k$-th discrete time interval.

Redefining the charge state of a supercapacitor as the ratio of the square of the voltage value in a certain state to the square of the voltage value in a fully charged state, the function of the no-load voltage of a supercapacitor with respect to its charge state is as follows:

$$U_{sc}(k) = \sqrt{U_{sc,N} \cdot SOC_{sc}(k)} \tag{3}$$

where, $U_{sc}(k)$ is the voltage of the supercapacitor under full charge; $SOC_{sc}(k)$ is the charged state of the ultracapacitor at the $k$ discrete time interval.

## Two-state variable based DPA

The concept of state variables pertains to variables that articulate the inherent state of a process at the outset of each stage within a DPA. Furthermore, the input state variable in each stage is contingent on the output state variable of the preceding stage. While the traditional DPA employs single-state variables to represent the process, such an approach imposes certain limitations in adequately capturing the intricacies of electricity consumption scheduling within the framework of a smart grid. To address this issue, our article proposes the use of dual-state variables to enhance the effectiveness of the DPA.

The grid energy consumption scheduling process is deemed a problem-solving process, which is segmented into n stages based on the time sequence of the demanded power during system operation. In the context of smart grid energy consumption scheduling, the state of charge of energy storage devices, namely, the battery and supercapacitor, serves as an indicator of the entire system's state. Therefore, the $SOC$ of battery and supercapacitor are taken as the state variables of the system $z_1(k)$ and $z_2(k)$, and the decision variable $j(k)$ is taken as the output power of battery $P_b(k)$, then the output power of supercapacitor can be determined as:

$$P_{sc}(k) = P_{req}(k) - u(k). \tag{4}$$

The discrete situation of the state equation of the smart grid energy storage system is shown in Eq. (5), where the state variables in the $k+1$-th phase are determined by the state variables and decision variables in the $k$-th phase.

$$\begin{cases} z_1(k) = f_1(z_1(k), u(k)) \\ z_2(k) = f_2(z_2(k), u(k)) \end{cases} \tag{5}$$

where $f_1$ and $f_2$ are the transfer functions for the state variables $z_1(k)$ and $z_2(k)$, respectively.

Equations (6) and (6) depict the present computation for the battery and supercapacitor, along with the state transfer equations.

$$\begin{cases} I_b(k) = \dfrac{U_b(k) - \sqrt{U_b^2(k) - 4R_b u(k)}}{2R_b} \\ z_1(k+1) = z_1(k) - \dfrac{I_b(k)\Delta t(k)}{Q_{b,N}} \end{cases} \tag{6}$$

$$\begin{cases} I_{sc}(k) = \dfrac{U_{sc}(k) - \sqrt{U_{sc}^2(k) - 4R_{sc}P_{sc}(k)}}{2R_{sc}} \\ z_2(k+1) = z_2(k) - \dfrac{P_{sc}(k)I_{sc}^2(k)R_{sc}}{\frac{1}{2}C_{sc,N}U_{sc,N}^2} \end{cases} \tag{7}$$

where, $\Delta t(k)$ indicates the duration of the $k$ phase; $Q_{b,N}$ is the rated capacity of the battery; $C_{sc,N}$ is the rated capacitance of the supercapacitor; and $U_{sc,N}$ is the rated voltage of the supercapacitor.

The expression of the optimal objective function for a certain state variable at the $n$-th stage is:

$$J(z_1(n), z_2(n)) = \min_{u(n)}(L(z_1(n), z_2(n), u(n))) \tag{8}$$

where $L$ represents the cost function, which is the loss value of the smart grid energy storage system.

The optimal objective function used to determine the optimal decision in the $k$-th phase, when the state variables are determined, is shown in Eq. (9). The meaning is: the total loss value of the grid energy storage system from the departure of the $k$-th state to the last state process.

$$J(z_1(k), z_2(k)) = \min_{u(k)}(L(z_1(k), z_2(k), u(k))) + J(z_1(k+1), z_2(k+1)) \tag{9}$$

where $z_1(k)$ and $z_2(k)$ are the state variables of smart grid, and $u(k)$ isthe decision variable.

## EXPERIMENT AND ANALYSIS

This article's experimental setup is bifurcated into two distinct segments. The first segment aims to substantiate the efficacy of the proposed LSTM-based electricity consumption prediction method that factors in environmental considerations. Specifically, in this section, we compare the performance of the time series model and the degree of fit of the prediction curve for the method considering environmental factors against the one that does not. The second segment of the experiment involves the evaluation of the overall smart grid energy scheduling system for energy conservation. To this end, we compare the performance of comparable algorithms using various indices to validate the efficacy of the DPA that incorporates the improved two-state variables proposed in this article. The experimental environment comprised an Intel(R) Xeon(R) CPR E5-2620 @ 2.40 GHz processor with 64G running memory and NVIDIA Tesla V100 GPU, whereas the software environment involved Pytorch 1.11.0 under the Ubuntu 20.0.4 operating system.

Huang et al. (2023), *PeerJ Comput. Sci.*, DOI 10.7717/peerj-cs.1482

## Electricity consumption prediction

The following are the evaluation metrics used to assess the LSTM approach that incorporates environmental factors:

(1)    MAPE (Mean absolute percentage error), sensitive to relative error, does not change due to global scaling of the target variable:

$$MAPE(y, \sim y\%) = \frac{1}{n_{samples}} \sum_{i=0}^{n_{amples}-1} \frac{|y_i - \sim y_i\%|}{\max(\varepsilon, |y_i|)} \qquad (10)$$

(2)    MSE (Mean squared error), the mean of the absolute squared errors of the predicted and true values:

$$MSE(y, \sim y\%) = \frac{1}{n_{samples}} \sum_{i=0}^{n_{samples}-1} (y_i - \sim y_i\%)^2 \qquad (11)$$

This experiment aims to validate the effectiveness of using LSTM models for electricity consumption prediction after integrating environmental factors. Specifically, the experiment involves training a model that combines environmental variables and comparing its performance against a model trained using only electricity consumption data as input. The comparison of the two models' performance in terms of evaluation metrics is presented in Table 1. Among them, only "ours" combines environmental variables, and the rest methods only use electricity consumption data for training. The results demonstrate that after combining environmental factors, the LSTM model's MAPE reduces by 0.553%, from 4.565% to 4.014%, compared to training with ordinary electricity consumption data. Additionally, after combining environmental factors, the LSTM model's MSE decreases from 0.2846 to 0.2645, a reduction of 0.0201. Furthermore, we compared the performance of other time series models, namely the recurrent neural network (RNN) and CNN-LSTM, against the LSTM model with environmental factors. The LSTM model with environmental factors outperformed both the RNN and CNN-LSTM models, with the latter being the second-best performer. These findings support the notion that incorporating environmental factors can enhance the time series forecasting model's electricity consumption prediction capabilities compared to using only raw data.

Figure 4 presents a comparison of the predicted electricity consumption using the LSTM method with environmental factors proposed in this article and the curve fit of the test set. The red part of the plot represents the prediction result of the training process, the blue part represents the prediction result of the test set, and the green curve is the original data. The plot illustrates that both the training and test data align closely with the curve of the original data set,. The curve is almost completely fitted to the training set, but there are slight errors in the fitting to the original data set. This finding demonstrates that the LSTM method combined with environmental factors accurately predicts the electricity consumption of a smart grid.

**Table 1** **Effectiveness of power consumption among different models.** The table presents results of the ablation experiments, where ten power generation modules were utilized for simulation, and displays the average energy consumption values over a specific period.

| Methods | MAPE (%) | MSE |
|---|---|---|
| RNN | 4.875 | 0.2947 |
| LSTM | 4.567 | 0.2846 |
| CNN-LSTM | 4.234 | 0.2798 |
| Ours | 4.014 | 0.2645 |

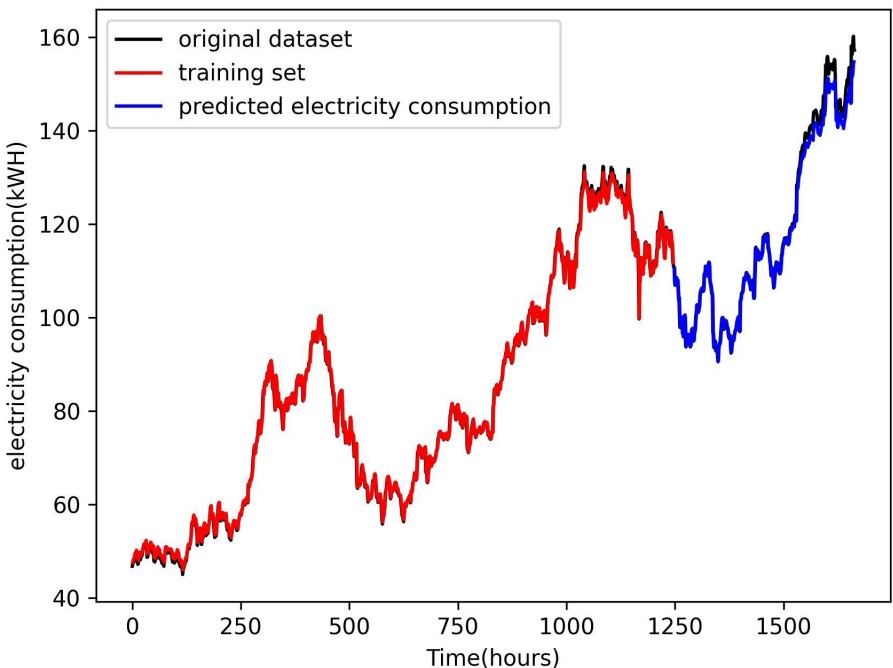

**Figure 4** **Comparison of power consumption forecast results.** The figure presents comparison of the predicted electricity consumption using the LSTM method with environmental factors proposed in this article and the curve fit of the test set. The red part of the plot represents the prediction result of the training process, the blue part represents the prediction result of the test set, and the green curve is the original data.

## Energy consumption scheduling

The experimentation was conducted using MATLAB (MathWorks, Inc., Natick, MA, USA), where a total of ten generation modules were simulated. Ablation experiments were devised to test the efficacy of the proposed smart grid energy consumption scheduling system's two components. Firstly, the dynamic programming scheduling algorithm was omitted, and secondly, the traditional DPA was used for scheduling optimization. Thirdly, the two-state variable improved DPA was employed, and fourthly, the two-state variable DPA was combined with the LSTM electricity consumption prediction method, taking into account environmental factors. Comparison experiments were also conducted to validate the overall system's effectiveness, where the particle swarm optimization algorithm, the

**Table 2   Ablation results.** The proposed two methods reduced the smart grid system's energy consumption to varying degrees.

| Methods | Energy consumption (mkJ) |
|---|---|
| DP | 64.3 |
| DP+LSTM | 58.1 |
| Ours | 49.5 |

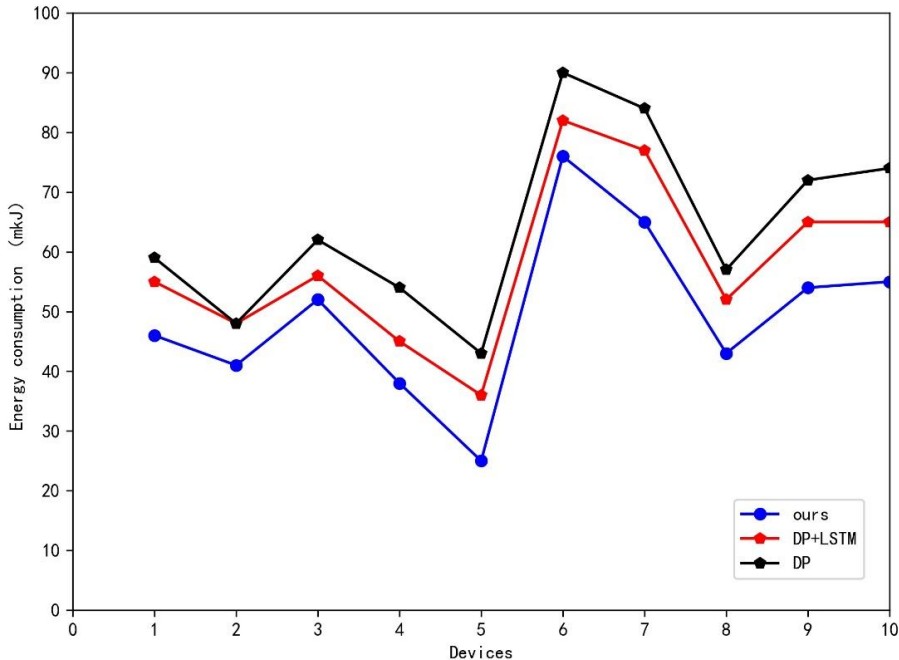

**Figure 5   Ablation experiment results.** The data shows that the presented method had a comparable effect on different power generation modules, reducing energy loss in varying degrees.

ant colony algorithm, and the random forest algorithm were compared to the same type of energy consumption scheduling algorithms.

Table 1 presents the results of the ablation experiments, where ten power generation modules were utilized for simulation, and Table 1 displays the average energy consumption values over a specific period. Furthermore, line graphs were plotted to showcase the different methods' performances on various generation modules. As shown in Table 2 and Fig. 5, the two methods proposed in this article reduced the smart grid system's energy consumption to varying degrees. The addition of the LSTM method with environmental factors proposed in this article reduced the mean energy consumption of ten generation modules by 6.2 mkJ compared to the traditional DPA. By combining the two-state variable DPA, the overall system saved 14.8 mkJ compared to the traditional DPA, amounting to approximately 23% of energy loss. The data in Fig. 5 shows that the method presented in this article had a comparable effect on different power generation modules, reducing energy loss in varying degrees.

**Table 3  Comparative experiment results.** The results of the comparison experiments are shown. The data presented here shows that our proposed method exhibits varying degrees of improvement when compared to the three machine learning algorithms typically used in energy scheduling: random forest, the ant colony algorithm (ACO) and the particle swarm algorithm (PSO).

| Methods | Energy consumption (mkJ) |
| --- | --- |
| Random Forest | 65.2 |
| ACO | 60.7 |
| PSO | 56.2 |
| Ours | 49.5 |

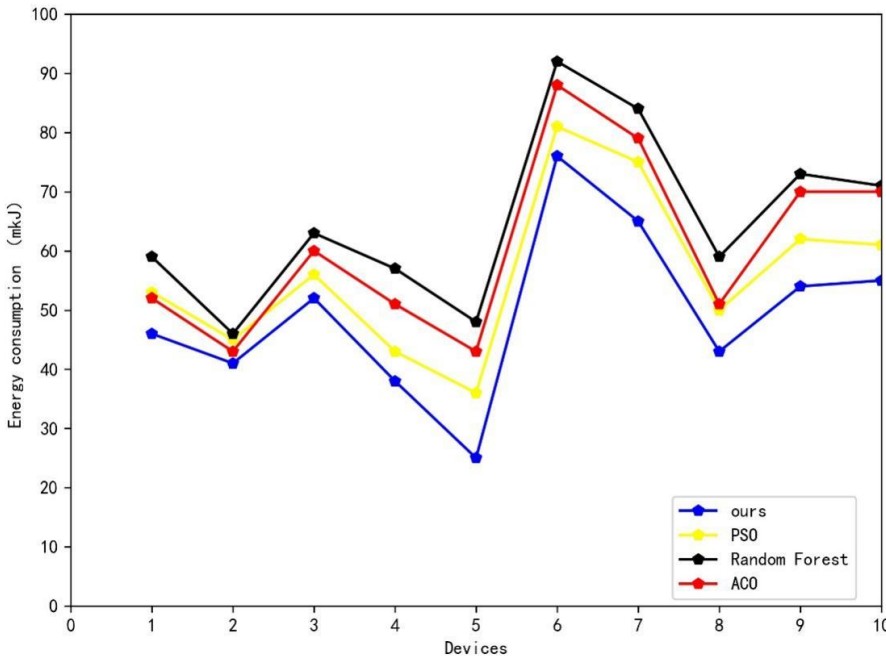

**Figure 6  Results of comparative experiment.** The figure depict the results of the comparison experiments.

Table 3 and Fig. 6 depict the results of the comparison experiments. The data presented in Table 3 shows that our proposed method exhibits varying degrees of improvement when compared to the three machine learning algorithms typically used in energy scheduling: random forest, the ant colony algorithm (ACO), and the particle swarm algorithm (PSO). Our method achieves an energy reduction of 15.7 mkJ on a simulation experiment of a power generation module, as compared to the worst-performing random forest algorithm. In comparison to the second-best performing PSO algorithm, our method achieves a positive improvement of 2.7 mkJ less. In summary, using the method proposed in this article for energy scheduling in smart grids can significantly reduce energy consumption across power-using devices of varying power levels.

## Discussion

Based on the results of the ablation experiments conducted on the time series model, the LSTM method incorporating environmental factors outperforms the original LSTM and RNN, and even surpasses the CNN-LSTM, which has been previously shown to outperform the LSTM. These findings highlight the significance of incorporating multiple factors in grid electricity consumption prediction. In terms of the overall scheduling algorithm, adding the LSTM incorporating environmental variables and the two-state variable DPA have varying degrees of performance improvement for the overall algorithm, and the effectiveness of both algorithms has been demonstrated in this article. The final comparison experiments confirm that the whole system can reduce energy consumption to varying degrees for different power generation modules, and the performance of energy consumption reduction is similar, with even more energy consumption reduction achieved in the lowest power module 5. These results underscore the universality and practical significance of the method presented in this article.

## CONCLUSION

In this article, we have presented a novel approach for energy consumption scheduling in smart grids by utilizing the LSTM algorithm and an improved DPA. Our approach incorporates environmental factors as input variables for power consumption prediction and introduces a dual-state variable to enhance the DPA's effectiveness. The results have demonstrated that the inclusion of environmental factors can lead to better fitting of real power consumption curves, and the improved DPA with dual-state variables can provide better energy consumption scheduling and energy savings for smart grids. The application of our method can effectively reduce energy consumption in the grid, thus contributing to energy conservation and environmental protection.

Future work can focus on incorporating more environmental factors, including not only natural factors but also factors related to grid equipment, into the grid electricity consumption prediction. Additionally, further improvements can be made to the dynamic programming process beyond the consideration of state variables alone. In addition, we plan to gather more comprehensive data and carry out feature extraction through different deep learning models to verify the generalization of the method.

## ACKNOWLEDGEMENTS

We would like to express our thanks to the Economic and Technological Research Institute of State Grid Fujian Electric Power Co., Ltd. for their help in this manuscript.

### Funding

The authors received no funding for this work.

## Competing Interests

All the authors are employed by Economic and Technological Research Institute of State Grid Fujian Electric Power Co., Ltd.

## Author Contributions

- Xiaoyu Huang conceived and designed the experiments, performed the experiments, analyzed the data, prepared figures and/or tables, authored or reviewed drafts of the article, and approved the final draft.
- Yubin Lin conceived and designed the experiments, analyzed the data, prepared figures and/or tables, and approved the final draft.
- Xiaofei Ruan performed the experiments, performed the computation work, prepared figures and/or tables, and approved the final draft.
- Jiyu Li analyzed the data, performed the computation work, authored or reviewed drafts of the article, and approved the final draft.
- Nuo Cheng analyzed the data, performed the computation work, authored or reviewed drafts of the article, and approved the final draft.

## Data Availability

The raw data and code are available in the Supplemental Files.

## Supplemental Information

Supplemental information for this article can be found online at http://dx.doi.org/10.7717/peerj-cs.1482#supplemental-information.

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
