# Peer review of "Smart grid energy scheduling based on improved dynamic programming algorithm and LSTM"

_PeerJ Computer Science, doi:10.7717/peerj-cs.1482_

## Round 0.1 · original submission · Major Revisions

Your paper needs couple of improvements, therefore please revise it carefully and resubmit for further consideration. Please also improve the language of the paper.

Reviewer 1 ·

Basic reporting

a. It is suggested that the keywords of the abstract of the paper can be increased to 5.
b. There is no specific introduction of relevant parameters in Formula 9 in this paper, so detailed elaboration is needed.
c. In the experiment and analysis part of the paper, the introduction to the experimental results in Figure 3 is written as Figure 4, please correct it.
d. Section 3 of the paper does not introduce the relevant definition of the LSTM model and the process of processing data. Please supplement accordingly.
f. DPA based on two state variables includes state variables of the k stage and decision variables. Compared with a single state, decision variables are added. What is the advantage of doing this.
g. In this paper, it is suggested to change the position of the two evaluation indicators, MAPE and MSE, and put them in Section 4.2.
h. The paper concludes with suggestions and more expectations and plans for future work.

Experimental design

e. How does the energy storage system development model of smart grid proposed in this paper apply the DPA and LSTM models proposed in this paper? Can you draw a full-text flow frame diagram to illustrate.

Validity of the findings

g. In this paper, it is suggested to change the position of the two evaluation indicators, MAPE and MSE, and put them in Section 4.2.

Additional comments

a. It is suggested that the keywords of the abstract of the paper can be increased to 5.
b. There is no specific introduction of relevant parameters in Formula 9 in this paper, so detailed elaboration is needed.
c. In the experiment and analysis part of the paper, the introduction to the experimental results in Figure 3 is written as Figure 4, please correct it.
d. Section 3 of the paper does not introduce the relevant definition of the LSTM model and the process of processing data. Please supplement accordingly.
e. How does the energy storage system development model of smart grid proposed in this paper apply the DPA and LSTM models proposed in this paper? Can you draw a full-text flow frame diagram to illustrate.
f. DPA based on two state variables includes state variables of the k stage and decision variables. Compared with a single state, decision variables are added. What is the advantage of doing this.
g. In this paper, it is suggested to change the position of the two evaluation indicators, MAPE and MSE, and put them in Section 4.2.
h. The paper concludes with suggestions and more expectations and plans for future work.

Reviewer 2 ·

Basic reporting

This paper introduces an enhanced dynamic programming algorithm (DPA), which uses two state variables to obtain the optimal power supply plan. In addition, combined with long and short term memory (LSTM) deep learning model, this paper seems to have proposed a method that significantly reduces energy consumption and is superior to similar power grid energy scheduling algorithms. This is crucial to build a smart grid and reduce energy consumption and emissions. Here are a few suggestions for improving the quality of the paper:
(1) The introduction of the paper introduces traditional methods and machine learning to solve the power system energy scheduling problem, which should be put in the introduction of related work.
(2) The contributions of this paper are not described in the relevant work section of this paper, and the work and contributions of the authors of this paper need to be added..
(3) There is no specific introduction of relevant parameters in Formula 3 in this paper, so it needs to be elaborated in detail.
(4) The introduction and explanation of experimental results in Table 2 in the experiment and analysis section of the paper is written as Table 1 in the paper, please correct it.
(5) The introduction should include some background on smart grids and energy scheduling.
(6) As for the improved DPA proposed in this paper, it is not explained in Section 3 where the improvement is compared with DPA. Please specify the improvement point.
(7) The figure in the experimental part of the paper does not correspond to the description of the experimental results. Please check and correct it carefully.
(8) The experiment and analysis section of the paper states that the LSTM method combined with environmental factors is better than the previous LSTM-related methods. Please give the specific experimental comparison.

Experimental design

(1)The figure in the experimental part of the paper does not correspond to the description of the experimental results. Please check and correct it carefully.
(2)The experiment and analysis section of the paper states that the LSTM method combined with environmental factors is better than the previous LSTM-related methods. Please give the specific experimental comparison.

Validity of the findings

no comments

---

## Round 0.2 · accepted · Accept

Thank you for your updated submission, consequent upon the recommendations of the experts, I'm pleased to inform you that your paper has been recommended for publication.
Thanks

Reviewer 1 ·

Basic reporting

Required adjustments have been made.

Experimental design

Required adjustments have been made.

Validity of the findings

Required adjustments have been made.

Reviewer 2 ·

Basic reporting

The given comments has been properly incorporated.

Experimental design

No comment.

Validity of the findings

no comment

Additional comments

The given changes has been properly made.